# Study on Wellbore Stability Evaluation Method of New Drilled Well in Old Reservoir

Shiyue Wang [1,*], Gaolong Liao [1], Zhenwei Zhang [2] and Xiaoyun Wang [2]

1   China National Offshore Oil Corporation Hainan Branch, Haikou 570311, China; liaogl2@cnooc.com.cn
2   College of Petroleum Engineering, Xi'an Shiyou University, Xi'an 710065, China;
    zhangzhenwei0831@163.com (Z.Z.); wxyun7@126.com (X.W.)
*   Correspondence: wangshy26@cnooc.com.cn

**Abstract:** The borehole wall stability of depleted reservoirs is a key problem that restricts the deeper potential exploitation of old oilfields. Based on these, according to the seepage and elastic mechanics theory of porous media, combined with laboratory rock mechanics experiments, the dynamic evaluation model of geomechanical parameters of the old reservoir is established, and on this basis, the evaluation method of wellbore stability of new wells in the old reservoir is established, and the changing laws and influencing factors of the wellbore stability of the old reservoir are quantitatively evaluated. The research shows that reservoir pressure, reservoir porosity, effective stress coefficient, strength, in situ magnitude, and direction were all changed due to long-term exploitation; both collapse and fracture pressure decreased, the loss risk of drilling fluid was increased, and safe drilling azimuth was changed. The results can guide reasonable choices such as well position, track and mud density, and strengthening sealing while drilling in order to ensure wellbore stability effectively.

**Keywords:** old reservoir; new drilled well; wellbore stability; evaluation method

## 1. Introduction

Depth exploration of old oilfields is one of the main directions for increasing production [1,2]. As the field continues to be developed over the years, production from old wells gradually declines, and the cost of oil and gas production rises year on year. In order to further explore the potential of old oilfields and fully release the capacity of old reservoirs, it is necessary to develop old reservoirs again (Figure 1, Reservoir A) or to exploit new reservoirs (Figure 1, Reservoir B) that have not been used at a deep layer below the old reservoirs. For old reservoirs, encrypted adjustment wells, etc., methods can be used to drill new horizontal wells (Figure 1, well B3) in the old reservoirs to obtain the remaining oil resources; for deeper unused new reservoirs, newly drilled adjustment wells (Figure 1, well B2) and sidetracks on old wells (Figure 1, well A is an old well, and B1 is a new borehole sidetrack on A) could be used in the new reservoirs.

However, no matter what method of exploration is used, it is inevitable that old reservoirs will have to be drilled through. With the continued development of the field over many years, significant pressure depletion typically occurs within older reservoirs (Figure 1, t1→t2), resulting in changes to reservoir ground stress, porosity, and mechanical properties. At this point, if it is still treated as the original reservoir, especially if it draws on the actual drilling of previous production wells near this reservoir, it may lead to unexpected complications such as wall collapse, well leakage, and stuck drilling during the drilling of the adjustment well [3–7]. Therefore, determining the effect of pressure depletion within old oil layers, optimising the borehole stability analysis model, and determining a reasonable mud density for new wells drilled in old oil layers play a vital role in the success of exploration and development in the middle and later stages of old oil fields. Ge H. [8], Addis M.A. [9], Liang H. [10], Shi M. [11], and Tan Q. [12] et al. established a model for calculating borehole stability in depleted reservoirs based on the study of the effect

of reservoir pressure depletion on ground stress. However, all of these research methods only consider changes in ground stress and do not take into account changes in reservoir plane pressure drop and mechanical properties, which makes them less applicable to the stability analysis of newly drilled well walls in old oil reservoirs. Based on this, this paper establishes a dynamic evaluation model of the geomechanical parameters of the old oil layer, on the basis of which an evaluation method for the stability of the newly drilled well in the old oil layer is established, and the change law and influencing factors of the well stability in the old oil layer are quantitatively evaluated, which has theoretical and practical significance for the safe and efficient drilling of wells in the old oil layer.

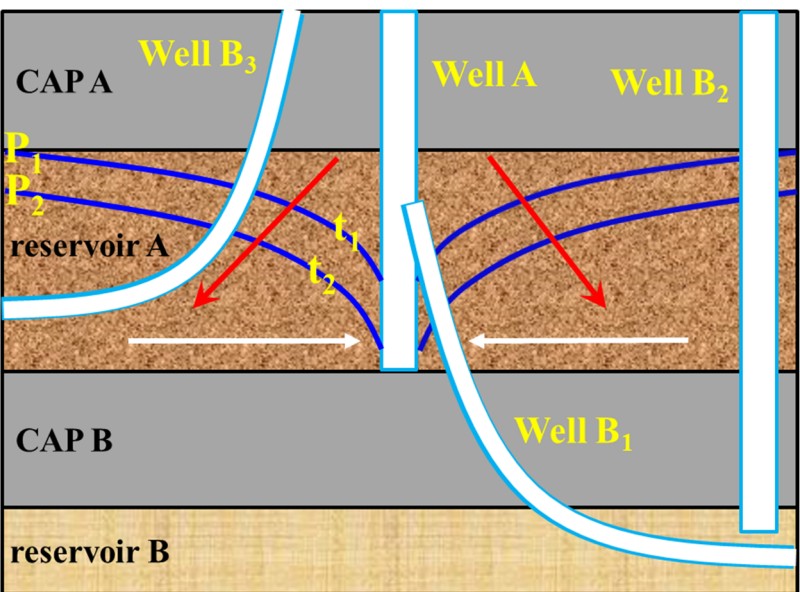

**Figure 1.** Newly drilled well in old reservoir schematic diagram.

## 2. Geomechanical Models

### 2.1. Pore Pressure

Affected by the oil and gas production of production wells, the reservoir pressure is depleted. The pressure at the production well, the pressure of the supply edge, and the location of the new well determine the pressure at the new well. The original direct use of reservoir pressure dynamic test data of production wells to predict the stability of new drilling wellbore ignores the difference in pressure depletion degree on the reservoir plane. Based on this, according to the theory of percolation mechanics, an analysis model of current reservoir pressure near new wells is established.

Assume that the supply boundary radius of reservoir is $R_e$, a borehole in the centre with a radius of $R_w$ and bottomhole pressure $p_w$; well B is the location of the new well as it passes through the old layer, at a distance r from the centre of well A. Assuming that the flow of reservoir fluid from the supply margin to production well A during development is a planar radial steady flow, the reservoir pressure at well B can be describe by Equation (1) [13]:

$$\frac{d^2P}{dr^2} + \frac{1}{r}\frac{dP}{dr} = 0 \tag{1}$$

The boundary conditions are:

$$\begin{cases} r = R_w, P = P_w \\ r = R_e, P = P_e \end{cases} \tag{2}$$

Solving the above definite solution problem yields the expression for reservoir pressure at newly drilled well B:

$$P(r) = P_w + \frac{P_e - P_w}{\ln \frac{R_e}{R_w}} \ln \frac{r}{R_w} \tag{3}$$

### 2.2. Porosity

In oil and gas exploitation, the reservoir pressure decays, and usually, the overlying rock stress remains constant, with a corresponding increase in the effective stress in the rock skeleton, resulting in most of the load from the upper rock layer being transferred to the rock skeleton, causing the compaction of the reservoir and a reduction in porosity [14]. This phenomenon was demonstrated experimentally by Geertsma [15]. Based on the effective stress principle and the definition of rock porosity and compression coefficient, the present-day porosity prediction model for old oil reservoirs under constant overlying pressure is obtained as follows:

$$\phi = \phi_0 - C_b(1 - \phi_0 - C_s/C_b)\mathrm{d}P \tag{4}$$

where $\phi_0$ is the initial porosity; $C_s$ is the rock skeleton compression factor, MPa$^{-1}$; $C_b$ is the rock compression factor, MPa$^{-1}$.

### 2.3. Effective Stress Coefficient

According to the effective stress principle, the deformation damage of porous rocks around a well is actually controlled by the effective stress, and the effective stress factor is the key parameter that determines the magnitude of the effective stress. Through an experimental study of the pore elasticity characteristics of sandstone [16], Ge et al. concluded that the effective stress coefficient of sandstone reservoirs is related to its own porosity, and he gave a prediction model for the effective stress coefficient based on the critical porosity with a correction for the surrounding pressure conditions, as shown in Equation (5):

$$a = \frac{1 - (1 - \phi/\phi_c)^n}{1 + \beta P_c} \tag{5}$$

where $a$ is the effective stress factor; $\phi_c$ is the critical porosity; $n$ is the stiffness factor; $\beta$ is the coefficient of influence of the surrounding pressure; $P_c$ is the surrounding pressure, MPa. According to the above equation, the effective stress factor of the reservoir is related to the porosity. Therefore, pressure depletion will cause a change in the effective stress factor.

### 2.4. Strength

Reservoir pressure decay induces changes in strength indirectly through changes in porosity. The porosity of the rock is an important parameter reflecting the degree of compactness; in general, the greater the porosity is, the lower the compressive strength is. There is usually a pattern of decreasing compressive strength of rocks with decreasing density, which is a concrete expression of the effect of increasing porosity on compressive strength. By studying the relationship between uniaxial compressive strength and porosity of three types of reservoir rocks, sandstone, shale, and tuff (Table 1), Chang [17] et al. found that the strength of all three types of rocks increased with decreasing porosity, and sandstone was the most sensitive to changes in porosity, followed by tuff, and shale was the weakest.

The above empirical formula is highly regional, and the number of cores required for the experiment is large. For this reason, this paper uses a combination of interpreting logging data from neighbouring wells and indoor tests to investigate the correlation between porosity and uniaxial compressive strength of the target reservoir, and combines the predicted results of the present-day porosity of the reservoir to achieve reservoir strength prediction in newly drilled wells.

**Table 1.** Relations between UCS and porosity in different rocks.

| Lithology | Relationship Formula | Applicable Conditions | |
|---|---|---|---|
| Sandstone | $UCS = 277\exp(-10\phi)$ | $2\text{MPa} < UCS < 360\text{MPa};$ | $0.002 < \phi < 0.33$ |
| Shale | $UCS = 1.001\phi^{-1.143}$ | $UCS > 79\text{MPa};$ | $\phi < 0.1$ |
| Limestone | $UCS = 135.9\exp(-4.8\phi)$ | $10\text{MPa} < UCS < 300\text{MPa};$ | $0 < \phi < 0.2$ |

The bulk density of the formation can be measured by density logging, according to the Wyllile formula [18]:

$$\rho_b = \phi\rho_f + (1-\phi)\rho_{ma} \tag{6}$$

Thus, it can be calculated that:

$$\phi = (\rho_{ma} - \rho_b)/(\rho_{ma} - \rho_f) \tag{7}$$

where $\rho_b$ is the density of the rock body, g/cm$^3$; $\rho_{ma}$ is the density of the rock skeleton, g/cm$^3$; $\rho_f$ is the bulk density of water, g/cm$^3$.

Logging sonic velocity and gamma are important indicators of constant rock strength parameters and mud content, and Deer and Miller established a mathematical relationship between the uniaxial compressive strength of sedimentary rocks and dynamic Young's modulus and mud content as follows [19]:

$$UCS = 0.0045 \cdot E_d(1 - V_{cl}) + 0.008E_d \cdot V_{cl} \tag{8}$$

In the above equation: $V_{cl}$ is the clay content; $E_d$ is dynamic Young's modulus, MPa (it can be calculated from logging sound waves and gamma, see literature [19] for the exact formula).

According to Equations (6)–(8), the porosity and uniaxial compressive strength of the sandstone are obtained by using the interpretation of logging data from neighbouring wells and indoor experiments, and the best curve fit of the relationship is obtained by regression analysis (Figure 2). The joint fitting equation and Equation (4) can be used to achieve the present-day strength prediction of the reservoir after pressure depletion:

$$UCS = 94.698e^{-6.087\phi} \tag{9}$$

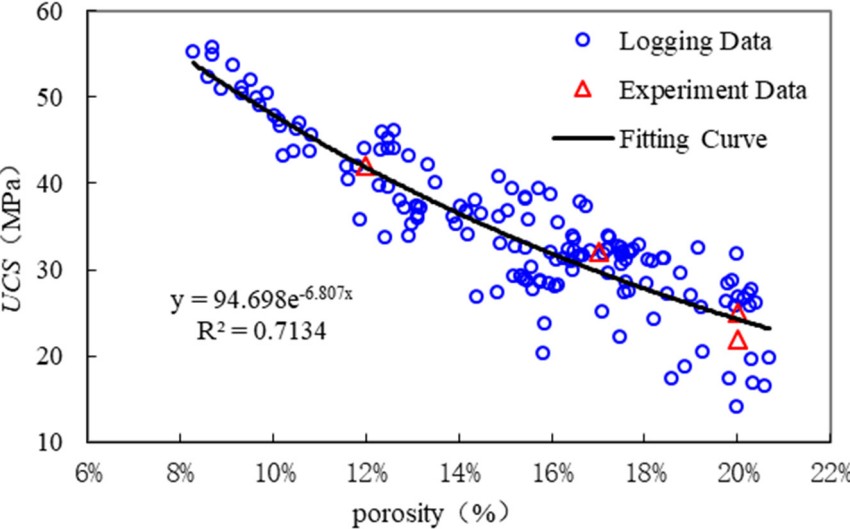

**Figure 2.** The relation between porosity and uniaxial compressive strength of sandstone.

*2.5. In Situ Stress*

2.5.1. In Situ Magnitude

Reservoir pore pressure decay will result in a change in the magnitude of horizontal stress [7]. Assuming that the reservoir is a homogeneous isotropic linear elastomer and that changes in pore pressure cause only vertical deformation of the formation, and considering the dynamic changes in the effective stress coefficient, the relationship between the amount of change in horizontal principal stress in the formation and the amount of change in pore pressure is:

$$\begin{cases} \sigma_{H1} = \sigma_H + \frac{(1-2\mu)(1-(1-\phi/\phi_c)^n)}{(1-\mu)(1+\beta P_c)}\Delta P_p \\ \sigma_{h1} = \sigma_h + \frac{(1-2\mu)(1-(1-\phi/\phi_c)^n)}{(1-\mu)(1+\beta P_c)}\Delta P_p \end{cases} \tag{10}$$

where $\sigma_H$ and $\sigma_h$, $\sigma_{H1}$ and $\sigma_{h1}$ are the horizontal maximum and minimum principal stresses before and after pressure decay, respectively, MPa; $\mu$ is the reservoir Poisson's ratio.

2.5.2. Stress Direction

For broken block fields, non-permeable faults often act as a barrier to pressure transfer. The different levels of reservoir mobilisation on either side of the fault result in a pressure difference near the fault, inducing additional ground stresses perpendicular to the fault strike at this location. If the fault orientation is not parallel to the original ground stress direction, the superposition will result in a deflection of the ground stress direction [20]. The angle of deflection in the direction of horizontal ground stress can be calculated by the following equation [21–24]:

$$\gamma = \frac{1}{2}\tan^{-1}\left[\frac{Aq\sin 2\theta}{1 + Aq\cos 2\theta}\right]; A = \frac{\alpha(1-2\mu)}{1-\mu}; q = \frac{-\Delta P_p}{\sigma_H - \sigma_h} \tag{11}$$

where $\theta$ is the angle between the direction of the original maximum horizontal ground stress and the strike of the fault.

## 3. Borehole Stability Evaluation Model

Based on the prediction model of reservoir pressure and changes in induced mechanical parameters, a model for calculating the stress state of rocks around the well under the synergistic effect of multiple factors is established, a suitable damage criterion is selected to judge the stability of the well wall rocks, and a model for analysis of the stability of the newly drilled well is established.

*3.1. Well Perimeter Stress*

3.1.1. Concentrated Stress on In Situ Stress and Drilling Fluids

When the borehole is drilled open, stress concentrations are generated, and the wellbore is subjected to a combination of three-way in situ ground stress and drilling fluid pressure. The component of ground stress at any borehole in a newly drilled well due to changes in the magnitude and orientation of the horizontal ground stress can be expressed by the following equation:

$$\begin{cases} \sigma_{xx} = \sigma_{H1}\cos^2\alpha\cos^2\beta_1 + \sigma_{h1}\cos^2\alpha\sin^2\beta_1 + \sigma_v\sin^2\alpha \\ \sigma_{yy} = \sigma_{H1}\sin^2\beta_1 + \sigma_{h1}\cos^2\beta_1 \\ \sigma_{zz} = \sigma_{H1}\sin^2\alpha\cos^2\beta_1 + \sigma_{h1}\sin^2\alpha\sin^2\beta_1 + \sigma_v\cos^2\alpha \\ \sigma_{xy} = -\sigma_{H1}\cos\alpha\cos\beta_1\sin\beta_1 + \sigma_{h1}\cos\alpha\cos\beta\sin\beta_1 \\ \sigma_{xz} = \sigma_{H1}\cos\alpha\sin\alpha\cos^2\beta_1 + \sigma_{h1}\cos\alpha\sin\alpha\sin^2\beta_1 - \sigma_v\cos\alpha\sin\alpha \\ \sigma_{yz} = -\sigma_{H1}\sin\alpha\cos\beta_1\sin\beta_1 + \sigma_{h1}\sin\alpha\cos\beta_1\sin\beta_1 \end{cases} \tag{12}$$

where $\sigma_v$, $\sigma_{H1}$, and $\sigma_{h1}$ are the overlying ground stress, present-day horizontal maximum, and minimum ground stress, respectively, MPa; $\alpha$ is the well slope angle; $\beta_1$ is the angle

between the borehole bearing and the $\sigma_{H1}$ bearing;$\sigma_{xx}$, $\sigma_{yy}$, $\sigma_{zz}$, $\sigma_{xy}$, $\sigma_{xz}$, and $\sigma_{yz}$ are the well stress components in the Cartesian coordinate system.

Superimposing the above stress components everywhere and the stresses induced by the drilling fluid column pressure at the perimeter of the well, the stress distribution on the wall of an arbitrary borehole is obtained as:

$$\begin{cases} \sigma_{r1} = P_i \\ \sigma_{\theta 1} = A\sigma_{h1} + B\sigma_{H1} + C\sigma_v - P_i \\ \sigma_{z1} = D\sigma_{h1} + E\sigma_{H1} + F\sigma_v \\ \sigma_{\theta z1} = G\sigma_{h1} + H\sigma_{H1} + J\sigma_v \\ \sigma_{r\theta 1} = \sigma_{rz} = 0 \end{cases} \tag{13}$$

In the equation:

$$\begin{cases} A = \cos\alpha\left[\cos\alpha(1 - 2\cos 2\theta)\sin^2\beta_1 + 2\sin 2\beta_1 \sin 2\theta\right] + (1 + 2\cos 2\theta)\cos^2\beta_1 \\ B = \cos\alpha\left[\cos\alpha(1 - 2\cos 2\theta)\cos^2\beta_1 - 2\sin 2\beta_1 \sin 2\theta\right] + (1 + 2\cos 2\theta)\sin^2\beta_1 \\ C = (1 - 2\cos 2\theta)\sin^2\alpha \\ D = \sin^2\beta_1\sin^2\alpha + 2\mu\sin 2\beta_1\cos\alpha\sin 2\theta + 2\mu\cos 2\theta\left(\cos^2\beta_1 - \sin^2\beta_1\cos^2\alpha\right) \\ E = \cos^2\beta_1\sin^2\alpha - 2\mu\sin 2\beta_1\cos\alpha\sin 2\theta + 2\mu\cos 2\theta\left(\sin^2\beta_1 - \cos^2\beta_1\cos^2\alpha\right) \\ F = \cos^2\alpha - 2\mu\sin^2\alpha\cos 2\theta \\ G = -\left(\sin 2\beta_1\sin\alpha\cos\theta + \sin^2\beta_1\sin 2\alpha\sin\theta\right) \\ H = \sin 2\beta_1\sin\alpha\cos\theta - \cos^2\beta_1\sin 2\alpha\sin\theta \\ J = \sin 2\alpha\sin\theta \end{cases}$$

### 3.1.2. Additional Stress on Drilling Fluid Seepage

After reservoir pressure failure, the pressure difference between the drilling fluid at the bottom of the well and the reservoir fluid increases, and the seepage of drilling fluid filtrate into the perimeter rock increases. Assuming that the fluid flow satisfies Darcy's law, and considering the pressure failure induced changes in reservoir porosity and effective stress coefficient, the calculation equation for seepage additional stress is obtained as follows:

$$\begin{cases} \sigma_{r2} = -\phi_1(P_i - P_{p1}) \\ \sigma_{\theta 2} = \left[\frac{\alpha_1(1 - 2\nu)}{(1 - \nu)} - \phi_1\right](P_i - P_{p1}) \\ \sigma_{z2} = \sigma_{\theta 2} \end{cases} \tag{14}$$

where $\phi_1$ is the present-day reservoir porosity; $P_{p1}$ is the present-day reservoir pressure, MPa; $\alpha_1$ is the present-day reservoir effective stress factor.

### 3.1.3. Total Stress Distribution around Borehole

According to the superposition principle, the total stress on the well wall during the drilling of the adjustment well is obtained by combining Equations (13) and (14):

$$\begin{cases} \sigma_r = P_i + \sigma_{r2} \\ \sigma_\theta = A\sigma_{h1} + B\sigma_{H1} + C\sigma_v - P_i + \sigma_{\theta 2} \\ \sigma_z = D\sigma_{h1} + E\sigma_{H1} + F\sigma_v + \sigma_{z2} \\ \sigma_{\theta z} = G\sigma_{h1} + H\sigma_{H1} + J\sigma_v \\ \sigma_{r\theta} = \sigma_{rz} = 0 \end{cases} \tag{15}$$

### 3.2. Borehole Stability Analysis Model

The two main modes of well destabilisation include collapse and fracture. The collapse of a borehole wall is mainly caused by the low pressure of drilling fluid at the bottom of the well, resulting in stresses around the well exceeding the compressive strength of the formation, and the critical pressure when a well wall collapses and destabilises is the collapse pressure. In conjunction with the reservoir present-day strength prediction model,

the Mohr–Coulomb criterion was used to determine whether the well wall was collapsing and destabilising:

$$(\sigma_{max} - a_1 P_{p1}) = (\sigma_{min} - a_1 P_{p1}) \cot^2\left(45° - \frac{\phi}{2}\right) + UCS_1 \tag{16}$$

where $\sigma_{max}$ and $\sigma_{min}$ are the maximum and minimum present-day principal stresses around the well, respectively, MPa; $\phi$ is the angle of internal friction of the rock, °; $UCS_1$ is the present-day reservoir compressive strength, MPa.

Well fracture is mainly due to the high pressure of drilling fluid at the bottom of the well, resulting in perimeter stress exceeding the tensile strength of the formation. The critical pressure at which a well fracture destabilisation occurs is the fracture pressure. Therefore, the criterion for well wall fracture destabilisation can be expressed by the following equation:

$$(\sigma_{min} - a_1 P_{p1}) = -S_{t1} \tag{17}$$

where $S_{t1}$ is the present-day reservoir tensile strength, which is generally 1/15–1/8 of the compressive strength, MPa.

From Equation (15), it can be seen that only the radial stress is the principal stress on the inclined shaft wall, and the other two principal stresses can be obtained by converting the plane stress as follows:

$$\sigma_{1,2} = \frac{\sigma_z + \sigma_\theta}{2} \pm \sqrt{\left[\frac{\sigma_z - \sigma_\theta}{2}\right]^2 + \sigma_{\theta z}^2} \tag{18}$$

Therefore, the maximum and minimum principal stresses on the borehole wall of any borehole are:

$$\begin{cases} \sigma_{max} = max(\sigma_1, \sigma_2, \sigma_r) \\ \sigma_{min} = min(\sigma_1, \sigma_2, \sigma_r) \end{cases} \tag{19}$$

Equations (16), (17) and (19) can be combined to obtain arbitrary borehole wall collapse and rupture pressures.

## 4. Changes in Reservoir Mechanical Parameters and Well Wall Stability

### 4.1. Reservoir Pressure Changes

Assuming a reservoir burial depth of 3000 m, a borehole radius of 215.9 mm, and a supply margin radius of 1 km, the pressure remains stable at 30 MPa during extraction. The variation pattern of reservoir pressure with the location of the newly drilled well and the degree of exploitation of the producing well is obtained according to Equation (3) (see Figure 3). Based on the results, it is clear that the closer the newly drilled well is to the location of the production well, the more severe the reservoir pressure depletion is, and that the difference between reservoir pressure and production well pressure is already large within approximately 20 times the borehole diameter near the production well, so there is a large error in the original direct use of dynamic test data of reservoir pressure from the production well to predict the well wall stability of the newly drilled well. In addition, there will be an overall trend towards lower pressures across the reservoir plane as the time spent on producing well increases.

### 4.2. Induced Mechanical Parameter Changes

Simulated parameters: reservoir initial pressure 30 MPa, reservoir initial porosity 25%, rock skeleton compression factor $1.5 \times 10^{-4}$ MPa$^{-1}$, rock compression factor $14 \times 10^{-4}$ MPa$^{-1}$, reservoir rock critical porosity 36%, stiffness factor 2.5, perimeter pressure 45 MPa, perimeter pressure influence factor 0.0014, initial horizontal maximum ground stress 54 MPa, initial horizontal minimum ground stress 45 MPa, Poisson's ratio 0.2, initial effective stress coefficient 0.893, original maximum horizontal ground stress direction N30°E, fault orientation positive N–S direction. The pattern of reservoir pressure decay-

induced changes in reservoir porosity, effective stress factor, strength, and magnitude and direction of ground stress are shown in Figures 4 and 5. According to the calculation results, it can be seen that as the pressure decays, most of the load from the upper rock layer is transferred to the rock skeleton, resulting in the compaction of the reservoir, which, in turn, causes a series of changes such as a decrease in reservoir porosity, a decrease in the effective stress factor, an increase in reservoir strength, a decrease in the horizontal maximum ground stress, a decrease in the horizontal minimum ground stress, and an orientation deflection.

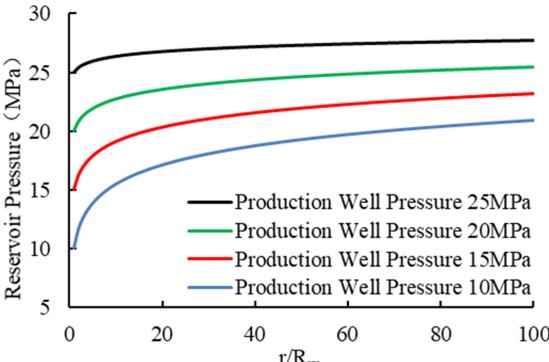

**Figure 3.** The evolution of reservoir pressure with adjustment wells' position and production wells' recovery level.

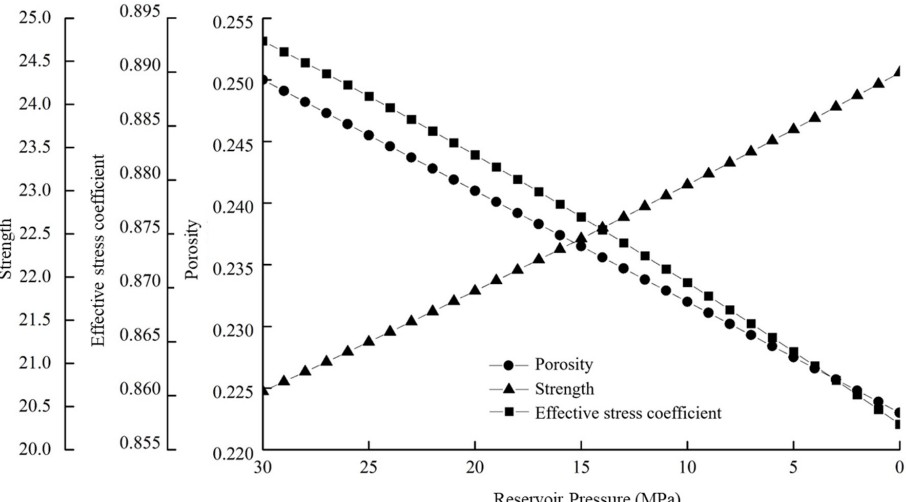

**Figure 4.** The evolution of induced mechanical parameter with pressure decline.

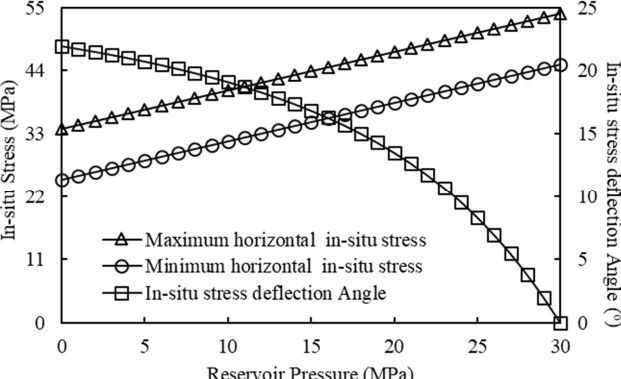

**Figure 5.** The evolution of in situ stress magnitude and direction with pressure decline.

### 4.3. Change in Stress State around the Well

Figure 6 gives the variation pattern of peri-well stress in straight wells with reservoir pressure depletion considering changes in effective stress factor, porosity, ground stress, and seepage added stress. Based on the calculations, it can be seen that as the reservoir pressure decays, the tangential stress around the well decreases, the vertical stress increases, and the radial stress decreases. The maximum well perimeter stresses occur at well perimeter angles of 90° and 270° when the reservoir pressure is not depleted. The tangential and radial stresses are the maximum and minimum principal stresses, respectively, while with pressure depletion, the maximum principal stresses around the well gradually change to vertical stresses, and according to the Mohr–Coulomb criterion, the shear damage of the rock around the well changes from being controlled by tangential-radial stresses to being controlled by vertical radial stresses.

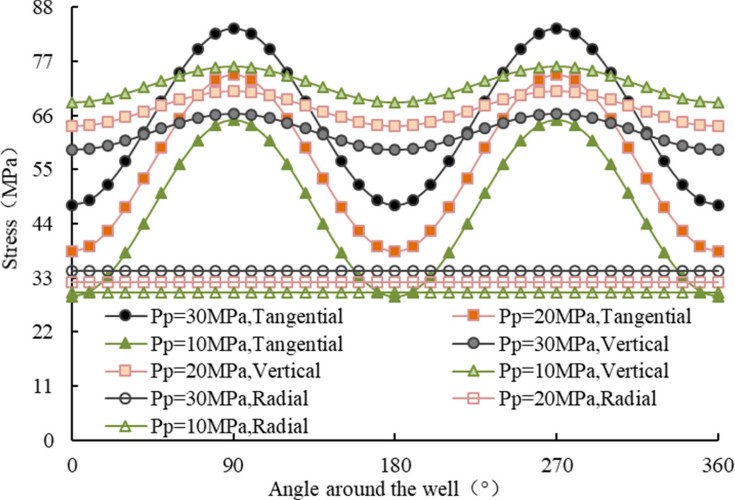

**Figure 6.** The evolution of total stress around well with pressure decline.

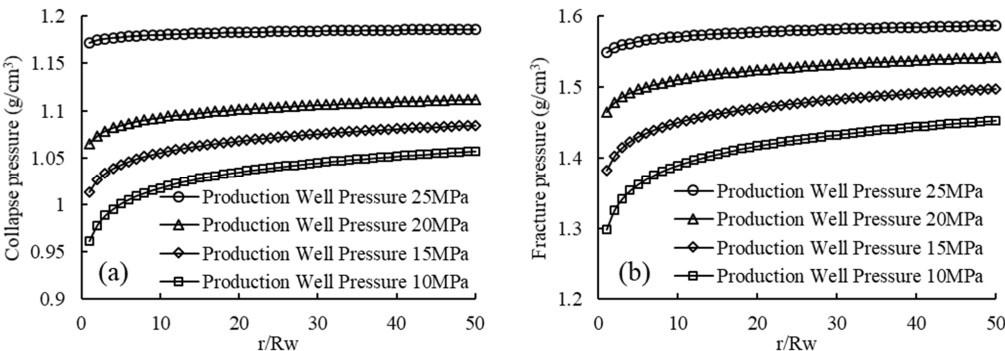

**Figure 7.** Reservoir collapse and fracture pressure change law of branch adjustment wells. (**a**): Collapse pressure evolution with different branch adjustment wells; (**b**): fracture pressure evolution with different branch adjustment wells.

### 4.4. Well Wall Stability Changes

4.4.1. Stability of Newly Drilled Well Walls in Depleted Reservoirs with Borehole Location and Degree of Pressure Depletion in Producing Wells

By considering the effects of non-uniform depletion of reservoir plane pressure, induced mechanical parameters, and seepage effects, the variation pattern of wall stability of newly drilled wells with the drilling location and the degree of pressure depletion in producing wells was obtained (Figure 7). Based on the results, it can be seen that the farther the newly drilled well is from the production well, the higher the collapse and rupture pressure will be under certain conditions of production well pressure; as the production well pressure decreases, the collapse and rupture pressure of newly drilled wells within a

certain range around the well will decrease, and the closer the well is to the production well, the greater the decrease will be. Therefore, the drilling of new wells requires a combination of predictions of well wall stability based on their drilling location and the degree of pressure depletion in producing wells.

### 4.4.2. Comparison of Results of Different Models for Calculating Changes in Well Wall Stability

Figure 8 gives the variation pattern of new drilling collapse and rupture pressures with reservoir pressure depletion as calculated by different models. Based on the results, it can be seen that the original model does not take into account the changes in mechanical parameters and seepage action during pressure decay, and the collapse pressure is low and the rupture pressure is high. The effect of seepage on well wall stability is becoming increasingly significant as reservoir pressure declines, and collapse pressure increases and rupture pressure decreases when seepage is considered. Combined consideration of mechanical parameters and seepage action changes compared to the original model also has the same collapse pressure increase and rupture pressure decrease. However, due to pressure decay caused by the reservoir porosity reduction, strength increase, effective stress coefficient, and other induced mechanical parameter changes, to a certain extent, this can slow down the change in seepage action and well wall stability. As a result, when the reservoir pressure decays to around 15 MPa, the magnitude of the change in collapse and rupture pressures slows down compared to when only seepage effects are considered.

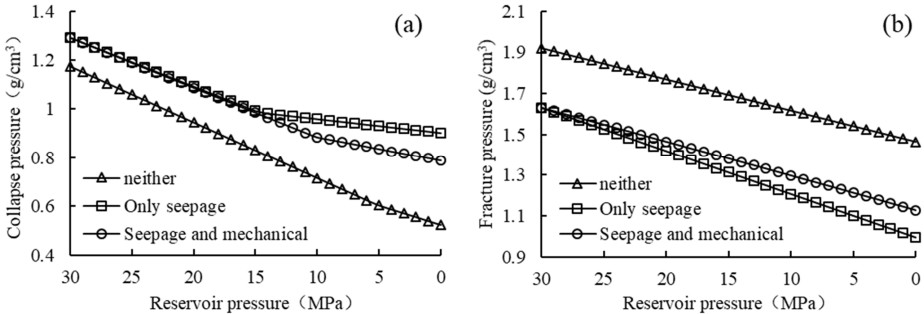

**Figure 8.** Reservoir fracture pressure change in contrast with different models. (**a**): Collapse pressure evolution with different branch adjustment wells; (**b**): fracture pressure evolution with different branch adjustment wells.

### 4.4.3. Patterns of Change in Wall Stability of Newly Drilled Wells with Arbitrary Trajectories

It is assumed that the original maximum horizontal ground stress direction is N30°E, that the fault is oriented in a positive N–S direction, and that the reservoir on one side of the fault has experienced pressure depletion through development. The pattern of new drilling collapse and rupture pressures with reservoir pressure depletion was calculated for arbitrary well slope and orientation (Figures 9 and 10). Based on the results of the calculations, it can be seen that as the reservoir pressure decreases, there is an overall decrease in the arbitrary borehole collapse and rupture pressure, and an increase in the risk of leakage during the drilling process: when reservoir pressure is not depleted (30 MPa), it is safest to drill new wells with a small slope towards N120°E, and the risk of destabilisation is greatest for new wells drilled with a large slope towards N30°E; pressure depletion to 20 MPa is safest when drilling small-slope wells towards N134°E, with the greatest risk of destabilisation when drilling large-slope new wells towards N44°E; pressure depletion to 10 MPa is safest when drilling small-slope wells towards N139°E, with the greatest risk to the wall of newly drilled wells with large slopes towards N49°E; pressure decay to the theoretical limit of 0 MPa is safest when drilling small-slope wells towards N142°E, and the risk of destabilisation is greatest when drilling large-slope new wells towards N52°E. Therefore, a reasonable design of the borehole trajectory and mud density based on the

degree of pressure depletion at the drilling location of a newly drilled well can effectively guarantee wall stability during the drilling process.

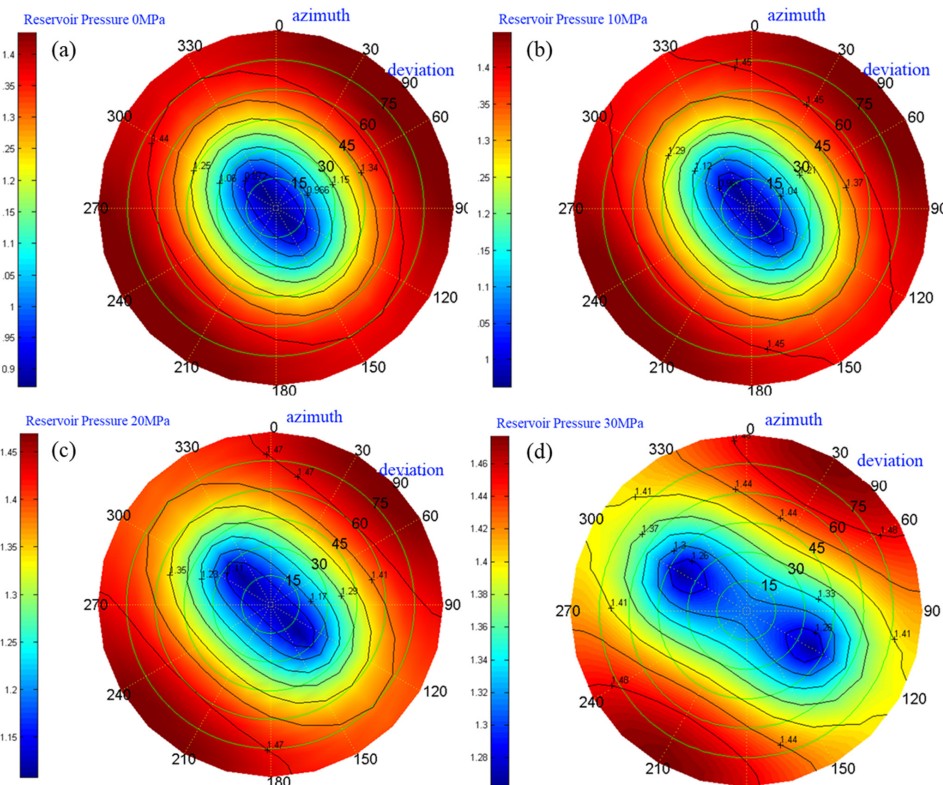

**Figure 9.** Reservoir collapse pressure change law of any well with pressure decline. (**a**–**d**) are represent reservoir pressure from 0 MPa to 30 MPa, respectively.

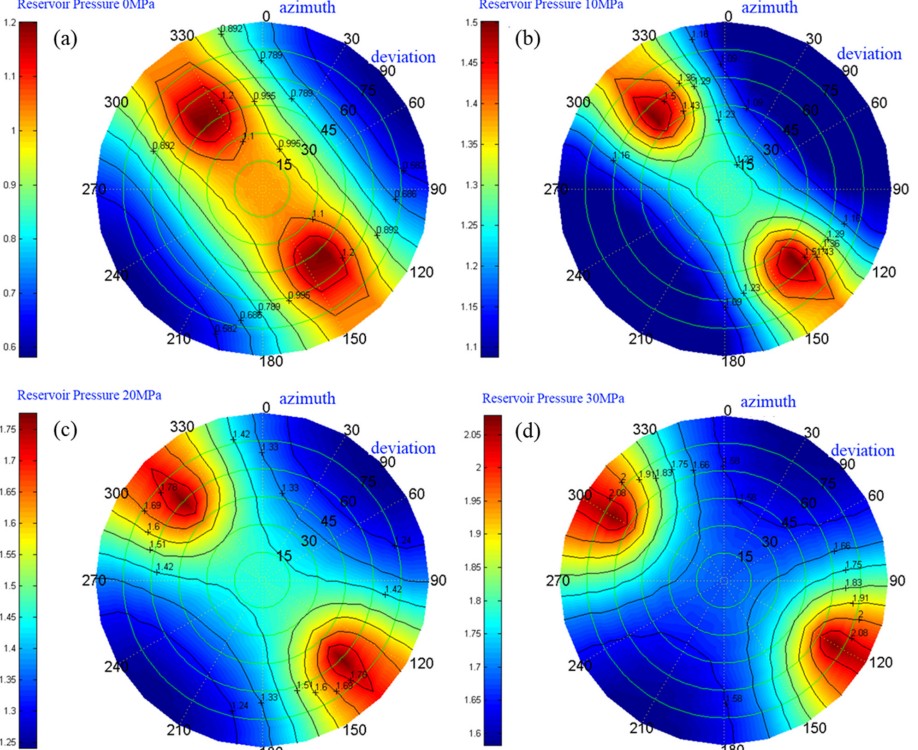

**Figure 10.** Reservoir fracture pressure change law of any well with pressure decline. (**a**–**d**) are represent reservoir pressure from 0 MPa to 30 MPa, respectively.

## 5. Conclusions

(1) Differences in the degree of pressure depletion in the reservoir plane around the well due to pressure depletion in production wells. The closer the location is to the producing well, the more severe the reservoir pressure depletion. The original direct use of dynamic reservoir pressure test data from producing wells to predict wall stability in newly drilled wells had large errors. As the reservoir pressure gradually decays, it results in a series of changes such as a reduction in reservoir porosity, a decrease in the effective stress factor, an increase in reservoir strength, a reduction in horizontal maximum ground stress, a reduction in horizontal minimum ground stress, and orientation deflection.

(2) Changes in the stability of newly drilled well walls are influenced by a combination of factors, including reservoir pressure, changes in induced formation properties, and changes in seepage action. The new method combines the effects of these actions and allows for more accurate guidance on safe slurry density design.

(3) Reservoir pressure depletion leads to an increased risk of leakage and changes in safe drilling orientation during the drilling process, which should be mainly prevented during the drilling process. The new drilling design can effectively ensure well wall stability through the reasonable design of the borehole trajectory, mud density, casing sealing, and strengthening the plugging with drilling.

**Author Contributions:** Conceptualisation, S.W.; Data curation, G.L.; Formal analysis, Z.Z.; Investigation, X.W. All authors have read and agreed to the published version of the manuscript.

**Funding:** This research received no external funding.

**Institutional Review Board Statement:** Not applicable.

**Informed Consent Statement:** Not applicable.

**Data Availability Statement:** Not applicable.

**Conflicts of Interest:** The authors declare no conflict of interest.

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
