# Peer review of "Study on Wellbore Stability Evaluation Method of New Drilled Well in Old Reservoir"

_processes, doi:10.3390/pr10071334_

Round 1

Reviewer 1 Report

This paper presents a method to evaluate the stresses developed in a wellbore which is redrilled. It combines elements from geomechanics and flow in porous media. The paper is well written and there is plenty of information provided to the reader. The language is fluent as well. I cannot yet recommend the paper to be published in Processes before the following two critical questions are addressed successfully.

a) The authors discuss on the case of a producing well which is redrilled. They consider production of reservoir fluids, hence the fact that the pressure regime exhibits a logarithm decay while approaching the wellbore. Simultaneously, the authors consider effects (such as seepage) of the drilling fluid running in the existing borehole (as is the case with Well B1 in Figure 1). How would it be possible that those two things happen at the same time? If the well is redrilled, the production would have been stopped, the production string would have been removed, the annulus would be filled with mud and fluid pressures in the reservoir under production would have equilibrated (no log distribution any more). This new equilibration pressure will, of course, be lower than that of the virgin reservoir (which gives rise to the idea presented in this work) but it will not vary in space (no production, no pressure difference according to Darcy’s law). Please comment on that very critical point and correct me if I have missed it.

b) A great variety of models has been combined in this work including models for porosity, rock strength, stress direction and failure models. Unfortunately, there has been no experimental data to validate that selection. Therefore, simulated results depend on the laws utilized for the development of the stress model. How high is that dependence expected to be and what is the anticipated effect of the use of alternative, still similar models to the quality (not to the exact values) of the obtained results? In simple words, how strictly dependent are your conclusions on the utilized methods?

Author Response

  1. Due to our describe in the manuscript not clear enough, a incorrect viewpoint was shown to readers. Actually, full process could divide into two steps, frist is development in old reservoir, in this stage, a darcy flow appearance around the well, and pore pressure around suffer a logarithm distribution; The second stage is the potential digging of an old reservoir or devlopment of a new deep reservoir, which requires the closure of a lateral well or the drilling of a new well to traverse the depleted reservoir. Because the drilling fluid pressure is usually higher than the formation pressure, the drilling fluid will seepage flow from the wellbore to the formation. Therefore, these two models are applicable to different stages and are not contradictory.
  2. A validation of each model is important, but due to the difference of rock property, in-situ stress distribution characteristic et al. between different reservoirs, the choice of model is also different. There is no public data can we find, making it difficult to verify. In this paper, we just focus on the well instability cause by sidetrack or new drilling well, a most popular model has been chosen to introduces one of the existing problems and shows that the phenomenon can not be ignored, and the porosity and strength models in practical use need to be selected according to regional experience.

Reviewer 2 Report

It is delighted to have the opportunity to review the proposed paper in the MDPI Processes Journal. This manuscript deals with one of the most interesting issues of well wall stability from the perspective of petroleum rock mechanics. The manuscript's structure is well organized; however, a few points should be considered to promote the manuscript quality and interest for the readers of the Processes journal:

11.   The manuscript initially highlights the need to develop well drilling in old oil and gas reservoirs, but the normality of the research is not well defined either in the abstract or in the introduction. It is recommended that this point be clearly considered.

22.  Abstract suffers from non-presentation of results and achievements of this research. Since the abstract of the article is actually known as an article showcase, sufficient focus on providing an attractive abstract will encourage readers to follow the topic and read the entire article.

33.    Although brief references to the issue of reservoir depletion are included in the text, the issue of reservoir depletion and changes in stress field balance in older reservoirs is very important. It is recommended to consider the effects of reservoir drainage and subsidence of underground layers as well as the general effects of these changes on the instability or destruction of new well walls. For this purpose, the following reference is suggested to enrich the content and references of the article:

A geomechanical approach to casing collapse prediction in oil and gas wells aided by machine learning." Journal of Petroleum Science and Engineering 196 (2021): 107811

44.    The creep effects of pressurized salt layers or their lateral displacements are also major threats to drilling in old and depleted oil fields. Therefore, it is strongly recommended that this issue be considered as an effective factor in the issue of well wall stability. The following reference will be valuable in explaining this issue and enriching the content:

"Shear modulus prediction of embedded pressurized salt layers and pinpointing zones at risk of casing collapse in oil and gas wells." Journal of Applied Geophysics 183 (2020): 104205.”

55. Providing research findings and discussing each of the findings, which is structured along with graph analysis, requires further explanation of the research findings. Authors need to work on enriching this section.

66. It is recommended that authors include at least two references from process journal articles in the text.

Author Response

  1. The normality of the research has been defined in the first paragraph.
  2. Abstract has been improved to highlight the results and achievements of this research.
  3. Reference has been added to paper.
  4. Reference has been added to paper.
  5. We have analyzed every figure in this paper, but because there are too many explanatory words, they are not reflected in the name of the picture, so they are put in the text before and after the picture.
  6. We newly cite two excellent papers which published on Processes in references 23 and 24.

Round 2

Reviewer 1 Report

No further comments after receiving the revised version of the manuscript.

Reviewer 2 Report

The authors have addressed well the concerns raised and the suggestions made have been heeded. This article is now recommended for publication in the Process Journal.